# Invadosome Formation by Lung Fibroblasts in Idiopathic Pulmonary Fibrosis

**DOI:** 10.3390/ijms24010499

**Published:** 2022-12-28

**Authors:** Mégane Lebel, Dominic O. Cliche, Martine Charbonneau, Damien Adam, Emmanuelle Brochiero, Claire M. Dubois, André M. Cantin

**Affiliations:** 1Respiratory Division, Department of Medicine, Université de Sherbrooke, Sherbrooke, Québec, QC J1H 5N4, Canada; 2Department of Immunology and Cell Biology, Université de Sherbrooke, Sherbrooke, Québec, QC J1H 5N4, Canada; 3Centre de Recherche du Centre Hospitalier de l’Université de Montréal (CRCHUM), Montréal, QC H2X 0A9, Canada; 4Department of Medicine, Université de Montréal, Montréal, QC H3T 1J4, Canada

**Keywords:** lung fibrosis, human lung fibroblast, nintedanib, bleomycin model, matrix degradation

## Abstract

Idiopathic pulmonary fibrosis (IPF) is characterized by abnormal fibroblast accumulation in the lung leading to extracellular matrix deposition and remodeling that compromise lung function. However, the mechanisms of interstitial invasion and remodeling by lung fibroblasts remain poorly understood. The invadosomes, initially described in cancer cells, consist of actin-based adhesive structures that coordinate with numerous other proteins to form a membrane protrusion capable of degrading the extracellular matrix to promote their invasive phenotype. In this regard, we hypothesized that invadosome formation may be increased in lung fibroblasts from patients with IPF. Public RNAseq datasets from control and IPF lung tissues were used to identify differentially expressed genes associated with invadosomes. Lung fibroblasts isolated from bleomycin-exposed mice and IPF patients were seeded with and without the two approved drugs for treating IPF, nintedanib or pirfenidone on fluorescent gelatin-coated coverslips for invadosome assays. Several matrix and invadosome-associated genes were increased in IPF tissues and in IPF fibroblastic foci. Invadosome formation was significantly increased in lung fibroblasts isolated from bleomycin-exposed mice and IPF patients. The degree of lung fibrosis found in IPF tissues correlated strongly with invadosome production by neighboring cells. Nintedanib suppressed IPF and PDGF-activated lung fibroblast invadosome formation, an event associated with inhibition of the PDGFR/PI3K/Akt pathway and TKS5 expression. Fibroblasts derived from IPF lung tissues express a pro-invadosomal phenotype, which correlates with the severity of fibrosis and is responsive to antifibrotic treatment.

## 1. Introduction

Idiopathic pulmonary fibrosis (IPF) is a chronic and progressive lung disease characterized by alveolar epithelial cell injury, aberrant epithelium repair, fibroblast accumulation and excessive deposition of extracellular matrix (ECM). Fibroblasts and myofibroblasts form fibroblastic foci, the accumulation of which is associated with disease progression and poor prognosis [1]. Pharmacological treatment options are currently limited to two antifibrotic drugs, nintedanib and pirfenidone. Although their cellular mechanisms of action remain to be fully elucidated, these antifibrotics have been shown to slow the progressive decline in pulmonary functions as measured by forced vital capacity (FVC) [2,3]. Originally developed as an anti-angiogenesis agent, nintedanib inhibits the receptor tyrosine kinases (RTK) VEGFR, FGFR and PDGFRα/β at the ATP-binding pocket site. Nintedanib also inhibits members of the Src-family of kinases, Src, Lyn and Lck [4]. Moreover, this molecule has been reported to inhibit PDGFR and the downstream signaling pathways Akt and ERK in the bleomycin-induced lung fibrosis mouse model [5].

Fibroblasts are known to promote the aberrant ECM remodeling in several respiratory diseases such as asthma, chronic obstructive pulmonary disease and particularly interstitial lung diseases including IPF [6]. in IPF. In addition to releasing excessive amounts of collagen and fibronectin, they express mediators of matrix degradation, such as the metalloproteinases MMP-2 and MMP-9 [7] and enzymes responsible for ECM crosslinking, resulting in matrix stiffening [8]. The stiff matrix further activates fibroblasts [9] and promotes matrix invasion [10,11]. Fibroblasts also release ECM-bound TGFβ through matrix contraction and MMP-mediated cleavage [12,13].

First identified in Src-transformed fibroblasts [14], invadosomes (known as invadopodia when specific to cancer cells) have been observed in highly invasive cancer cells [15]. Briefly, the process of invadosome formation consists of an integrin-based matrix adhesion step that enables the cells to probe the ECM. Specific proteins, including TKS5, cortactin, N-WASP, AFAP110, supervilin (SVIL), CD44 and ARP2/3, act cooperatively to initiate actin polymerization. TKS5 binds to the phosphatidylinositol PI(3,4)P2 found at the membrane and acts as a scaffold protein for the polymerization complex. The colocalization of cortactin or TKS5 with f-actin is commonly used to identify invadosomes in cells [16,17]. The assembly of an actin bundle with microtubules and intermediate filaments such as vimentin provides a mechanical force to push the cell membrane protrusion into the ECM [18]. Local invasion of the matrix by invadosomes is facilitated by matrix degradation involving proteolytic enzymes, such as the serine protease FAP-α and the metalloproteinases MMP14 (MT1-MMP), MMP-2, MMP-9 and ADAM12 [19]. The formation of invadosomes is involved in the ECM remodeling observed in many malignant tumors and non-cancer cells such as macrophages, dendritic cells, osteoclasts and synoviocytes [20,21,22]. Invadosome formation is promoted in response to a variety of growth factors, signaling pathways and environmental cues [23]. In this regard, we recently reported the critical role of the PDGFR/PI3K/Akt signaling pathway in invadosome formation by arthritic fibroblast-like synoviocytes [24].

Although the involvement of fibroblasts in the dysregulation of ECM homeostasis is an important event in the development of fibrosis, little is known about the key players involved in this process. We hypothesized that invadosomes, which are known to drive ECM remodeling in cancer, might be assembled by IPF fibroblasts and that these structures could be targeted by first-line treatments. To address this question, fibroblasts were isolated from human and murine fibrotic lungs and assessed for their ability to form invadosomes. In this study, we report the presence of an invadosome-associated gene signature in IPF lungs and fibroblastic foci. Accordingly, IPF lung fibroblasts have an increased capacity to form invadosomes, an event that correlates with the severity of fibrosis observed in the patients’ lungs. Furthermore, nintedanib and pirfenidone decreased invadosome formation by IPF fibroblasts, with nintedanib concomitantly inhibiting TKS5 expression and the PDGFR/Akt axis. These data suggest that invadosomes are associated with IPF pathogenesis and reveal an additional mode of action for some of the current first-line treatments.

## 2. Results

### 2.1. Several Key Genes of Invadosome Formation Are Upregulated in IPF Lung Tissue Samples and in Fibroblastic Foci Areas

Given the possible presence and involvement of invadosomes in IPF disease, we first asked whether IPF lungs are characterized by a pro-invadosomal gene expression signature. For this, we used the public gene expression dataset GSE32537 and performed differential expression analysis between control and IPF subsets. The upregulation of fibrosis-related genes in IPF samples was used to validate the dataset. The results indicated that fibroblast-associated genes COL1A1, COL3A1, ACTA2, FN1, HAS2 and TGFB1 were strongly increased in IPF samples compared with the controls (Figure 1A). Next, using the same cohort, a panel of genes known to be associated with invadosome formation [17,25,26] was analyzed. The expression of FAP, ADAM12, SYNJ2, MMP2, VIM, AFAP1, SH3PXD2A (TKS5), SVIL and CD44 was significantly increased in IPF lungs (Figure 1B). To further investigate whether fibroblastic cells expressed increased levels of invadosome-related genes, we performed differential expression analysis between control alveolar septae, IPF alveolar septae and IPF fibroblastic foci with the GSE19500 dataset. The expression of SYNJ2, SH3PXD2B, FAP, SVIL, ADAM12, SH3PXD2A, SLC9A1 MMP2, MMP14 and AFAP1 was strongly increased in IPF fibroblastic foci compared to non-fibrotic IPF alveolar septae (Figure 1C). Overall, IPF lung tissues and fibroblastic foci express more invadosomal markers than healthy tissues, suggesting an increased presence of invadosomes in IPF lung tissues.

### 2.2. Invadosome Formation Is Increased in Lung Fibroblasts Isolated from IPF Patients and Fibrotic Mice

To assess the ability of IPF fibroblasts to form ECM-degrading invadosomes, we isolated lung fibroblasts from healthy and IPF lung tissues, and the cells were subjected to invadosome assays. Fluorescent images show that IPF fibroblasts form more invadosomes and create larger areas of degraded matrix compared with fibroblasts from healthy donors. The areas of degraded matrix found 20 h after cell seeding often appear beside the cells, suggesting fibroblast motility on the gelatin matrix (Figure 2A). Specifically, invadosome-producing cells are 1.8 times more abundant in IPF fibroblasts than healthy fibroblasts (Figure 2B), and this cellular behavior was accompanied by a 2.4-fold greater capacity to degrade matrix (Figure 2C). Of note, the capacity of IPF fibroblasts to form invadosomes was not associated with fibronectin expression (Appendix A) or with FVC clinical measures or smoking status (Appendix A). Next, cortactin and TKS5 proteins are essential to produce invadosomes. By confocal microscopy, the colocalization of f-actin and cortactin as well as f-actin and TKS5 (Figure 2D) specifically confirmed the presence of invadosomal structures in IPF fibroblasts. The number f-actin and cortactin positive puncta were 1.8 times higher in IPF fibroblasts (Figure 2E). In addition, the mRNA level of TKS5 is increased 2.1-fold in IPF fibroblasts cultures (Figure 2F) and is positively correlated with the ability of fibroblasts to form invadosomes (Figure 2G). Accordingly, lung tissue immunohistochemistry revealed that cytoplasmic TKS5 is intensified in IPF lungs compared to healthy lungs (Figure 2H). Knockdown of TKS5 in IPF fibroblasts reduced the ability of cells to produce invadosomes, confirming the involvement of TKS5 (Appendix A).

To determine whether invadosome formation is also present in a model of induced fibrosis, we next studied lung fibroblasts isolated from mice, 28 days after exposure to bleomycin. Actively degrading cells are 1.7 times more frequent in bleomycin fibroblasts (Appendix A), and these cells have a 2.6-fold greater capacity to degrade matrix (Appendix A). Moreover, fibroblasts from bleomycin mice produce two times more invadosomal structures as measured by f-actin and cortactin clusters (Appendix A).

### 2.3. Fibroblast Invadosome Formation Correlates Positively with the Collagen Content of Neighbouring Tissue

To define whether fibroblasts from collagen-rich regions of the lung have an increased capacity to form invadosomes, tissues from apical and basal regions of the lung were collected for each IPF patient and processed for histology. Then, high-resolution images of the tissues were taken following Masson’s trichrome staining to quantify collagen. Histologically, these tissues consisted of a heterogeneous distribution of normal tissues as well as immature and mature fibrotic areas, alveolar inflammation, fibroblastic foci and dense areas with collagen deposition, which is entirely consistent with the diagnosis of IPF. In Figure 3A, tissues from the IPF-01 patient show predominant fibrosis at the bases of the lung (collagen 39%) compared to the apex (collagen 24%), which is consistent with a usual interstitial pneumonia (UIP) pattern. Four out of five patients have a greater amount of collagen in the basal region of their lungs compared to the apical region (Figure 3B). Interestingly, fibroblasts derived from the lung bases of these patients, also showed an increased capacity to form invadosomes when compared with fibroblasts derived from the upper lung zones. Furthermore, the only patient with a greater amount of fibrosis in the apical sample had increased invadosome formation in apical-derived fibroblasts. This observation suggested that the potential of fibroblasts to form invadosomes may be related to the amount of collagen present in their vicinity (Figure 3C). To further explore the potential correlation between collagen density and fibroblast invadosome formation within the same lung area, we examined 17 tissue samples from 12 IPF patients representing a broad range of collagen content. A strong positive linear correlation was found between the presence of collagen and the formation of invadosomes (Figure 3D). To further assess the in vivo relevance of invadosomes in collagen-rich samples and fibroblastic foci, we used the GSE32537 and GSE169500 datasets to correlate the expression of TKS5 (SH3PXD2A) with collagen I (COL1A1) in lung tissues from IPF patients. Analysis of 119 IPF lung samples indicated a significant and strong correlation between type 1 collagen and TKS5 expression (Figure 3E). Similar results were observed with IPF fibroblastic foci samples regarding TKS5 and collagen I (Figure 3F). Altogether, these results suggest that invadosome production is intensified in fibroblasts from collagen-rich regions of IPF lungs.

### 2.4. Nintedanib and Pirfenidone Inhibit Invadosome Formation by Lung Fibroblasts from IPF Patients

Given that nintedanib and pirfenidone, the two first-line IPF treatments, were shown to slow the rate of disease progression, it was of interest to investigate their effect on invadosome formation. Images of fibroblasts from an IPF patient illustrate the efficacy of nintedanib to impair invadosome-induced gelatin degradation (Figure 4A). Nintedanib markedly decreases the percentage of invadosome-forming cells from 7 IPF patients to the level of healthy fibroblasts (22% ± 2) (Figure 4B). The degraded area was also decreased by 60% and 73% using 0.2 µM and 0.5 µM nintedanib, respectively (Figure 4C). Furthermore, f-actin and cortactin puncta counts demonstrate that nintedanib reduces the formation of invadosomal structures by 39% (Figure 4D and Appendix A). Pirfenidone at 5 mM also reduces the percentage of invadosome-forming IPF fibroblasts to the level of healthy fibroblasts (22% ± 2) (Figure 4E).

### 2.5. Nintedanib Reduces the Expression of TKS5 and p-Akt in IPF Fibroblasts

Among the receptor tyrosine kinases (RTK) targeted by nintedanib, PDGFRα/β receptors are activated during IPF due to the strong production of PDGF by alveolar macrophages [27]. Using healthy donor fibroblasts, we found that PDGF-BB induces by 1.6-fold TKS5 mRNA expression (Figure 5A) and by 1.9 times the capacity of cells to form invadosomes (Figure 5B), which was completely blocked by nintedanib, suggesting that PDGFR activation is sufficient to promote invadosome formation. Similarly, when IPF fibroblasts are stimulated with PDGF-BB, TKS5 mRNA expression is increased 1.4 times (Figure 5C). Nintedanib reduces the expression of TKS5 at the mRNA level (Figure 5C) as well as at the protein level (Figure 5D) in unstimulated IPF fibroblasts. Using the same condition, expression levels of MMP2, MMP14 and ADAM12 metalloproteinases remained unchanged, and the pro-fibrotic genes COL1A1 and CTGF were significantly decreased, confirming the efficacy of nintedanib (Appendix A). Given that PDGFR mainly acts through the PI3K/Akt signaling as well as non-receptor TKs including Src, we sought to determine the effect of nintedanib on the Src/*p*-cortactin and PI3k/Akt signaling pathways. Nintedanib had no significant effect on Src phosphorylation at the Y416 activation site as well as cortactin phosphorylation at Y421, an Src phosphorylation site [28] (Appendix A). However, immunoblots performed on IPF fibroblasts reveal that 0.5 µM of nintedanib significantly decreases the phosphorylation of Akt (Figure 5E). Inhibition of Akt with triciribine and MK-2206 reduced the frequency of invadosomes in IPF fibroblasts (Figure 5F). Altogether, these results suggest that nintedanib impairs invadosome formation likely by disrupting the PDGFR/PI3k/Akt pathway and by downregulating the expression of TKS5.

## 3. Discussion

IPF is characterized by aberrant interstitial remodeling and abnormal ECM deposition in the lung interstitium. Fibroblastic foci (FF) comprised of activated fibroblasts and myofibroblasts are a histologic hallmark of IPF and strongly correlate with disease progression and mortality [29]. Cells comprising FF likely originate from mesenchymal progenitor cells, EMT, circulating fibrocytes, pericytes and resident fibroblasts [30,31,32] and have the capacity to migrate, invade and remodel areas of alveolar damage [33,34,35]. Invadosomal structures characteristically present in cancer cells are known to facilitate metastatic dissemination through pericellular ECM degradation. The identification of such specialized cellular structures and their potential involvement in interstitial remodeling of the IPF lung is lacking. Other actin-based structures have been found in lung fibroblasts, such as focal adhesions [11] and filopodia [36]; however, these structures lack the proteolytic activity characteristic of invadosomes [26]. In the current study, we report for the first time that IPF fibroblasts have an increased capacity to form invadosomes, which correlates with the severity of lung fibrosis and is blocked by the antifibrotic drugs nintedanib and pirfenidone. These observations suggest that IPF fibroblasts can form invadosomes and provide new insights into the mechanism by which fibroblasts may contribute to IPF physiopathology.

TKS5, an essential component of invadosomes, acts as a scaffold for the actin polymerization complex and takes part in MMP trafficking and ROS-mediated MMP activation [16]. Here, TKS5 (*SH3PXD2A*) expression levels were found to be elevated in tissues, fibroblastic foci and cells derived from patients with IPF, suggesting its involvement in the disease. Using a gelatin degradation assay and a colocalization assay with TKS5, cortactin and f-actin, we identified functional invadosomes in lung fibroblasts derived from IPF patients and bleomycin-exposed mice. Healthy lung fibroblasts could produce invadosome structures and degraded gelatin matrix, but these features were robustly increased in fibroblasts isolated from fibrotic lungs.

MMPs with gelatinase activity are augmented in lung samples, BALF and plasma derived from patients with IPF [37]. In addition, increased expression and activation of MMPs was shown to contribute to fibrocyte migration [38] and lung fibroblast invasion [39], two events associated with ECM remodeling. The degradation of ECM by invadosomes is mediated by MMPs recruitment and secretion at the tip of the protrusion, where MMP14 can activate pro-MMP-2 and pro-MMP-9 [40]. Our RNAseq data analysis showed an upregulation of *MMP14, MMP2* and *ADAM12* in the fibroblastic foci, which was consistent with the ability of isolated IPF fibroblasts to degrade ECM in invadosome assays. TGFβ and multiple growth factors are bound to the ECM in a latent form and can be released and activated by MMPs and ADAMs [13,41]. It is therefore possible that the marked increase in invadosome formation observed in IPF lung-derived fibroblasts promotes ECM remodeling through activation of TGFβ and/or other growth factors via pericellular secretion and activation of proteases. Interestingly, several invadosome inducers are upregulated in lung fibrosis, such as TGFβ, PDGF and LPA [42]. These factors are notably involved in various cellular events associated with the fibrotic response such as fibroblast proliferation, migration and differentiation into myofibroblasts, as well as ECM remodeling [43]. These pro-fibrotic factors may therefore contribute to establish an invadosomal phenotype in IPF fibroblasts. It would be of interest to evaluate, in the future, the effect of TGFβ and LPA on the production of invadosomes by lung fibroblasts.

Invadosomes have been largely studied in vitro, but growing evidence in different organisms indicates their role in vivo in both physiological conditions and malignancies [26,44]. To further explore the potential relevance of invadosomes in lung fibrosis, 17 IPF lung tissue samples were obtained to measure collagen content and invadosome formation from fibroblasts isolated from the same specimen. Interestingly, fibroblasts with the highest capacity to form invadosomes originated from areas of the lung tissue with the most severe fibrosis. Accordingly, in IPF lung tissues and fibroblastic foci, RNAseq analysis revealed that samples with high level of collagen I were also those expressing increased levels of the invadosome marker TKS5. Previous works reported that dense fibrillar type I collagen was a strong inducer of invadosome production by cancer cells and human fibroblasts [45]. During lung fibrosis, substantial changes take place in the ECM composition and organization, resulting in increased rigidity [8]. Loss of lung compliance due to tissue stiffening is clinically manifested by traction bronchiectasis and restrictive ventilatory defects, such as a decreased FVC and an increased FEV1/FVC ratio. At the cellular level, the stiff matrix is known to increase invasion and migration of lung fibroblasts [10,46]. In liver fibrosis and cancer cell invasion, matrix stiffness clearly acts as an inducer of invadosomal activity [47]. Collectively, these studies suggest that the marked increase in invadosome formation by fibroblasts derived from areas of severe fibrosis found in this work may have been promoted by the rigid matrix present in these densely fibrotic portions of the lung.

We observed that nintedanib can achieve ~50% inhibition of invadosome formation at a lower concentration (0.5 µM) than pirfenidone (5.0 mM). These concentrations, which are commonly used for in vitro studies, correspond to, respectively, ~7 fold and ~60 fold above the concentrations found in human plasma [48,49]. Consequently, nintedanib was selected for further mechanistic investigation. Nintedanib targets many receptor tyrosine kinases (RTKs) and non-RTKs, the latter including members of the Src family [4] that are known to participate in the initiation step of invadosome formation [28]. In this study, RNAseq data showed similar levels of Src (SRC) and cortactin (*CTTN*) expression between the control and IPF tissues. Immunodetection of the Src Y416 phosphorylation site and the Src-mediated cortactin Y421 phosphorylation site also suggested that the activity of these proteins was not modulated by nintedanib used at a concentration sufficient to block invadosome formation. Therefore, nintedanib seemed to inhibit invadosome formation through an Src-independent mechanism.

Among the RTKs targeted by nintedanib, inhibition of PDGFRβ receptor was shown to markedly attenuate bleomycin-induced lung fibrosis [50]. Our group found that PDGF-BB induces the formation of invadosomes in fibroblast-like synoviocytes through the PI3K/Akt pathway. Moreover, in rheumatoid arthritis fibroblast-like synoviocytes, inhibition of PDGFR efficiently blocked invadosome formation [23]. Consistent with this work, we found that PDGF-BB induced the formation of invadosomes and the expression of TKS5 in healthy lung fibroblasts, all of which could be prevented with nintedanib. In IPF fibroblasts, the inhibition of invadosome and TKS5 by nintedanib was concomitant with reduced Akt phosphorylation. Specific inhibition of Akt with triciribine or MK-2206 impaired invadosome formation, suggesting that this process was dependent on the PI3K/Akt pathway, which is likely affected by nintedanib. The downregulation of PI3K decreases the production of PI(3,4,5)P3 and PI(3,4)P2. Interestingly, the most significantly overexpressed gene found in IPF fibroblastic foci from the RNAseq analysis was SYNJ2, which encodes a phosphatase that produces PI(3,4)P2 [51]. The PI(3,4)P2 synthesis is a prerequisite for TKS5 membrane attachment prior to invadosome formation [52]. Therefore, nintedanib could inhibit invadosomes by indirectly impairing TKS5 recruitment. Although this potential mechanism requires further investigation, the fact that nintedanib prevented invadosome formation represents a novel mode of action for this antifibrotic.

Given the role of invadosomes in ECM remodeling, invasion and migration, the observation of an invadosomal phenotype in IPF lung fibroblasts suggests a contribution of invadosomes to the pathogenesis of IPF, the details of which remain to be defined. To further study the role of invadosomes in lung fibrogenesis, the bleomycin mouse model could be used, since invadosomes were more numerous in fibroblasts isolated from lungs of mice with bleomycin-induced fibrosis.

In conclusion, this study provides evidence suggesting that invadosome formation is a cellular mechanism by which IPF fibroblasts can promote interstitial remodeling of the lung and perpetuate pulmonary fibrosis. Given that the ability of IPF lung-derived fibroblasts to form invadosomes is strongly associated with fibrotic ECM, the capacity of fibroblasts to form invadosomes is likely to be relevant to numerous diseases with ECM remodeling and a fibrotic component. Despite the availably of two treatments that slow the rate of fibrosis progression, IPF remains a fatal lung disease for which effective treatments are needed. Further studies will be required to define whether the mechanisms leading to the increased capacity of IPF fibroblasts to form invadosomes could become relevant therapeutic targets in IPF.

## 4. Materials and Methods

### 4.1. Lung Fibroblast Isolation and Culture

After lung transplantation, human lung tissue samples from healthy donors and IPF patients were obtained from the Respiratory Cell and Tissue Biobank of CRCHUM, affiliated with the Quebec Respiratory Health research Network (biobanque.ca) (Table 1) after obtaining patient informed written consent prior enrolment (protocol CER CHUM #08.063) and approval of the research study (#2018–2437) by the U. Sherbrooke Institutional Review Board. Lung samples were minced and incubated in Dulbecco’s modified Eagle’s medium (DMEM) containing DNase 1 (50 µg/mL) and Liberase TL (0.25 mg/mL). The enzymatic digestion was stopped by EDTA (100 mM). The cells were passed through a 70 µm strainer and pelleted by centrifugation. Isolated cells were cultured at 37 °C with 5% CO_2_ in DMEM supplemented with 15% fetal bovine serum, penicillin, streptomycin and glutamine. Additional normal human lung fibroblasts (NHLF) # CC-2512 and IPF disease human lung fibroblasts (DHLF-IPF) # CC-7231 were purchased from Lonza and cultured under the same conditions. The purity of fibroblast cultures was analyzed by flow cytometry (Appendix A). Lung fibroblasts from healthy and 28-day post-bleomycin challenge mice were isolated and cultured in a similar manner (data supplement). Human and murine cells were used between passages 3 and 7. Reagents, immunoblotting, mRNA expression analysis (Table 2) and public gene expression analysis from GSE32537 and GSE169500 are further described in the data supplement.

### 4.2. Invadosome Formation Assessment

Coverslips were coated with Oregon green gelatin and prepared as previously described [53]. Lung fibroblasts were seeded on gelatin-coated coverslips without FBS. After a 5–40 h period, fibroblasts were fixed with PFA, permeabilized with saponin and blocked with BSA. Actin and nuclei were stained with phalloidin-Texas red and DAPI, respectively. Invadosomes were identified as actin-rich dots near areas of degraded gelatin using an epifluorescent microscope (Carl Zeiss Inc, Thornwood, NY, USA) at a magnification of 40×. Three hundred cells were counted per coverslip to determine the percentage of invadosome-forming cells. The areas of gelatin degradation per cell were measured with ImagePro software (Media Cybernetics, Rockville, MD, USA). To quantify the number of invadosome structures per cell, fibroblasts were stained with anti-cortactin antibody, phalloidin-Texas red and DAPI. Clusters of cortactin and actin micrographs from 20–25 cells were acquired using a scanning confocal microscope LSM Olympus FV1000 spectral (Olympus, Tokyo, Japan) with a 60× magnification.

### 4.3. Lung Histology and Collagen Quantification

Lung tissues were fixed with 10% formalin before paraffin embedding. Tissue sections of 4 µm were stained with Masson’s trichrome by the Histology Platform at the Université de Sherbrooke. TKS5 (NBP1-90454) immunohistochemical staining was detected using diaminobenzidine and counter-stained with Harris hematoxylin. Microscope slides were scanned at 20× and 40× magnifications with the Hamamatsu NanoZoomer 2.0 RS slide scanner (Hamamatsu Photonics, Bridgewater, NJ, USA). Collagen quantification was performed as described by Chen et al. [54]. The percentage of collagen-positive areas was measured for each patient specimen on two separate samples (~0.5 cm^3^) and on two non-serial sections.

### 4.4. Statistical Analysis

GraphPad Prism 8.01 software (GraphPad Software Inc. La Jolla, CA, USA) was used to perform the statistical analysis. Each individual value in the histograms corresponds to a single donor. Significance between two groups of non-normally distributed data was measured with an unpaired Mann–Whitney test. A Kruskal¬¬–Wallis test followed by Dunn’s multiple comparisons were used to compare more than two groups. Correlation strength was measured with a two-tailed Pearson’s test. Results are presented as mean ± SEM and *p* values were identified as * *p* < 0.05, ** *p* < 0.01, *** *p* < 0.001, **** *p* < 0.0001.

## Figures and Tables

**Figure 1 ijms-24-00499-f001:**
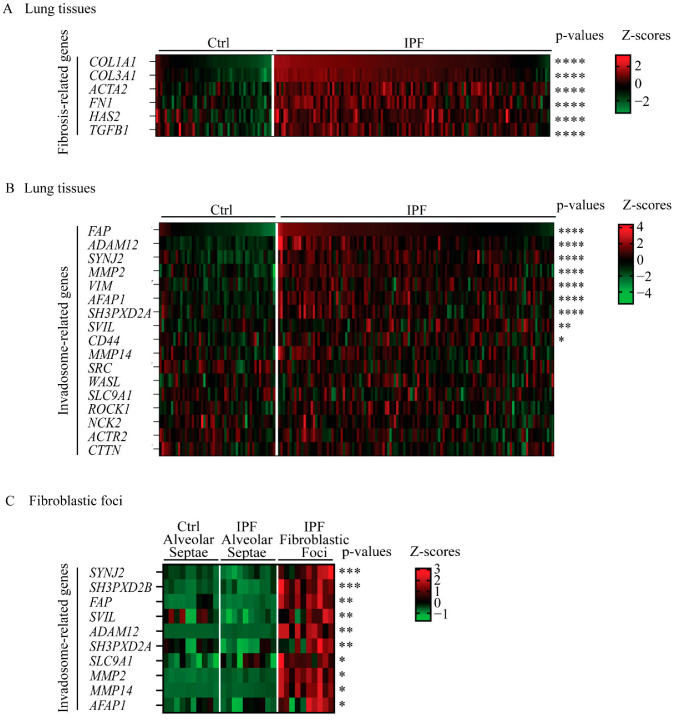
Several invadosome-associated genes are increased in lungs from IPF patients. Heatmaps of differential expression analysis using public datasets. Expression levels are presented as Z-scores. Significance of each gene is shown on the left with Benjamini–Hochberg false discovery rate adjusted *p*-values. Gene expression in lung tissue from GSE32537 of (**A**) 6 fibrosis-associated genes and (**B**) 17 genes encoding proteins involved in invadosome formation. Ctrl (*n* = 50) and IPF (*n* = 119). (**C**) Gene expression from GSE169500 of 10 genes involved in invadosome formation. Control alveolar septae (*n* = 10), IPF alveolar septae (*n* = 10) and IPF fibroblastic foci (*n* = 10). * *p* < 0.05, ** *p* < 0.01, *** *p* < 0.001, **** *p* < 0.0001.

**Figure 2 ijms-24-00499-f002:**
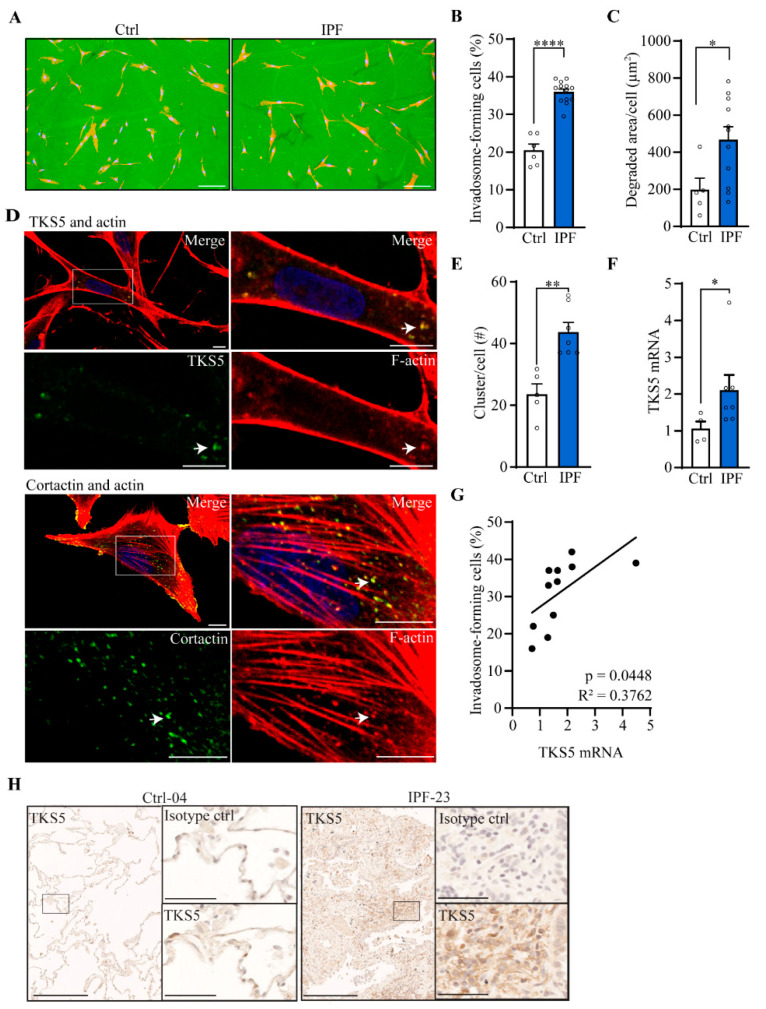
Invadosome formation is increased in lung fibroblasts isolated from IPF patients. Invadosome formation was assessed in fibroblasts from healthy and IPF lung tissues. Each value in the histograms corresponds to a primary culture from one healthy or one IPF individual. (**A**) Representative images of control and IPF human fibroblasts plated on fluorescent gelatin showing nuclei (blue) and f-actin (red). The invadosome-mediated degradation areas appear as black holes in the green gelatin in close proximity to the cells. Scale bar = 100 µm. (**B**) The percentage of cells forming invadosomes (ctrl: *n* = 6, IPF: *n* = 13) and (**C**) the areas of degraded gelatin per cell were quantified (ctrl: *n* = 5, IPF: *n* = 11). (**D**) Confocal micrographs of IPF fibroblasts stained for f-actin (red), nucleus (blue) and TKS5 (upper image) or cortactin (lower image) in green. An invadosome-rich region is zoomed in and presented on the right. Scale bar = 10 µm. (**E**) The number of invadosomal structures identified by the colocalization of cortactin and actin was counted per cell (ctrl: *n* = 5, IPF: *n* = 7). (**F**) mRNA level of TKS5 was measured (ctrl: *n* = 4, IPF: *n* = 7). (**G**) The levels of TKS5 mRNA expression were correlated with the percentage of invadosome-forming fibroblasts (*n* = 11). (**H**) Representative immunohistochemistry staining of TKS5 in healthy (*n* = 3) and IPF (*n* = 4) lungs. Scale bar = 250 µm. A region of interest is zoomed in and presented on the right with corresponding isotype control. Scale bar = 50 µm. * *p* < 0.05, ** *p* < 0.01, **** *p* < 0.0001.

**Figure 3 ijms-24-00499-f003:**
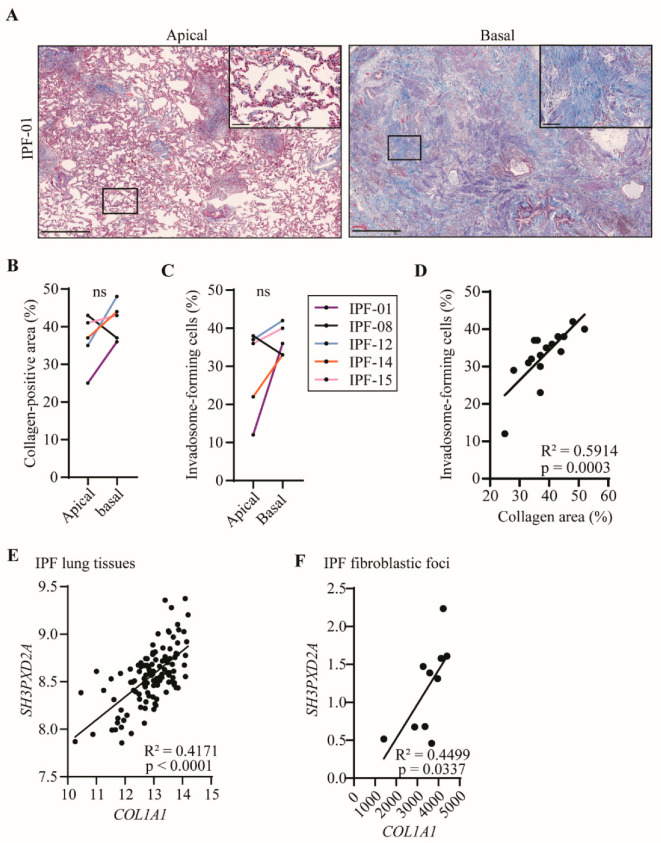
Invadosomes are found where collagen is heavily produced. (**A**) Images of apical and basal lung tissue sections stained for collagen with Masson’s trichrome from the same IPF patient showing different levels of fibrosis maturity. Scale bar = 1000 µm or 100 µm (zoom). For five patients, apical and basal lung samples were collected and (**B**) percentage of collagen-positive area as well as (**C**) percentage of fibroblast-forming invadosomes were determined. (**D**) Percentage of collagen-positive areas in lung tissues *versus* percentage of invadosome formation (*n* = 17). TKS5 (*SH3PXD2A*) relative gene expression was correlated with *COL1A1* expression in (**E**) IPF lung tissue (*n* = 119) and in (F) IPF fibroblastic foci (*n* = 10) using GSE32537 and GSE169500 cohorts, respectively.

**Figure 4 ijms-24-00499-f004:**
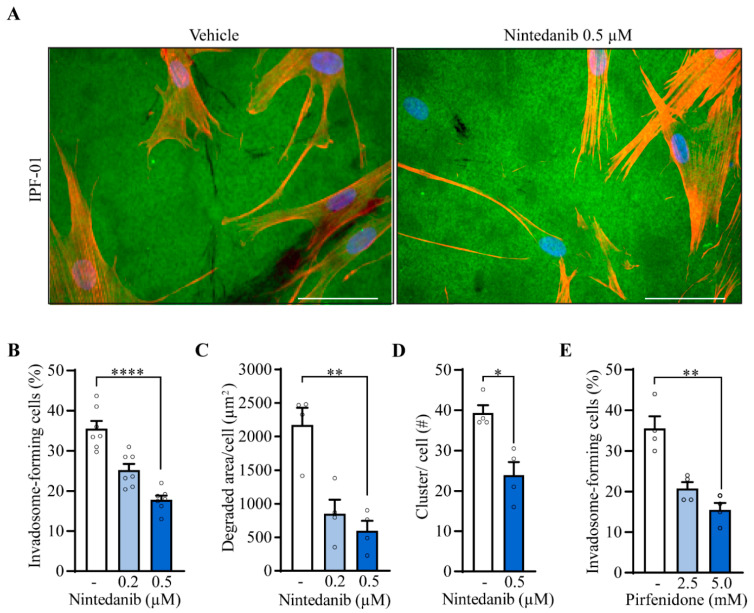
IPF treatments impair invadosome formation and function. IPF fibroblasts were incubated in the presence of nintedanib (*n* = 7). (**A**) Representative images of fibroblasts from the same patient plated on fluorescent gelatin (green) and stained for nuclei (blue) and f-actin (red). Scale bar = 50 µm. (**B**) The percentage of cell-forming invadosomes and (**C**) the surface degraded by fibroblasts (*n* = 4) were calculated. (**D**) Cortactin and f-actin clusters were counted (*n* = 4). (**E**) The percentage of invadosome-forming fibroblasts was determined in the presence of pirfenidone (*n* = 4). * *p* < 0.05, ** *p* < 0.01, **** *p* < 0.0001.

**Figure 5 ijms-24-00499-f005:**
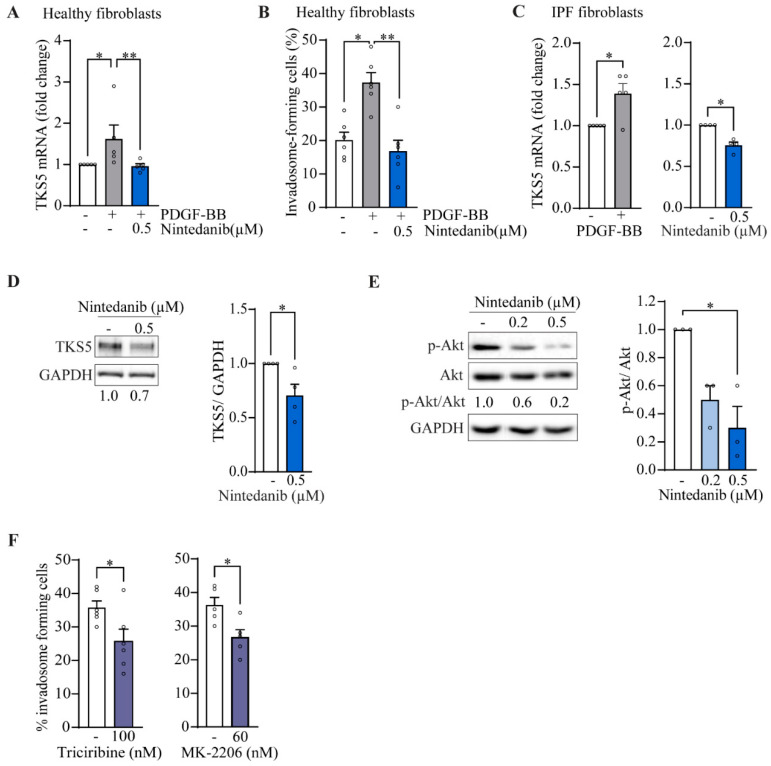
Nintedanib inhibits *p*-Akt and TKS5 in IPF fibroblasts. Healthy lung fibroblasts were cultured in the presence of PDGF-BB 25 ng/mL +/− nintedanib. (**A**) The mRNA levels of TKS5 (*n* = 5) and (**B**) the percentage of invadosome-forming cells (*n* = 6) were measured. IPF fibroblasts were cultured in the presence of PDGF-BB 12.5 ng/mL or nintedanib 0.2 µM and 0.5 µM. (**C**) mRNA levels of TKS5 were determined (*n* = 4−5) and expressed in folds relative to untreated cells. Immunoblot of (**D**) TKS5 and (**E**) phosphorylated Akt (S473) with corresponding densitometric analysis relative to untreated (*n* = 4). (**F**) The percentage of invadosome-forming cells in IPF fibroblasts treated with the Akt inhibitors triciribine (*n* = 6) and MK-2206 (*n* = 5). * *p* < 0.05, ** *p* < 0.01.

**Table 1 ijms-24-00499-t001:** Participant demographics.

Criteria	Healthy Donors (*n* = 8)	IPF Patients (*n* = 16)
Age (mean years ± SD)	56 ± 18	64 ± 8
Sex		
Male (%)	75	81
Female (%)	25	19
Smoking status		
Ever smoker (%)	37.5	56
Never smoker (%)	0	25
Unknown (%)	62.5	19
Years with IPF disease (mean ± SD)	*n*/a *	5.1 ± 3.1
IPF treatment before transplantation		
Nintedanib (%)	0	0
Pirfenidone (%)	0	12.5
FVC (% mean ± SD)	*n*/a *	41 ± 12

* *n*/a: not applicable or not available.

**Table 2 ijms-24-00499-t002:** Primer sequence used for qPCR.

Target Gene	Primer Sequences
** *SH3PXD2A* ** **(TKS5)**	Forward: 5′- TGC CAA GAA GGA GAT CAG CC-3′ Reverse: 5′-TGG AGG TCT TGT CCG TAG GT-3′
** *RPL13* **	Forward: 5′-CTC AAG GTC GTG CGT CTG- 3′ Reverse: 5′-TGG CTT TCT CTT TCC TCT TCT C-3′
** *COL1A1* **	Forward: 5′-AAG AGG AAG GCC AAG TCG AG-3′ Reverse: 5′-CAC ACG TCT CGG TCA TGG TA-3′
** *CTGF* **	Forward: 5′-AAT GCT GCG AGG AGT GGG T -3′ Reverse: 5′- CGG CTC TAA TCA TAG TTG GGT CT-3′
** *MMP2* **	Forward: 5′-GGC ACC CAT TTA CAC CTA CA -3′ Reverse: 5′-CCA AGG TCA ATG TCA GGA GAG -3′
** *ADAM12* **	Forward: 5′- TCT CAT TGC CAG CAG TTT CAC -3′ Reverse: 5′- CGT GTA ATT TCG AGC GAG GG -3′

## Data Availability

Gene expression matrix from GSE32537 dataset (https://www.ncbi.nlm.nih.gov/bioproject/PRJNA154433) and GSE169500 dataset (https://www.ncbi.nlm.nih.gov/bioproject/?term=GSE169500) were downloaded on 4 February 2022 and 11 May 2022 respectively from Gene Expression Omnibus.

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
