# Peer review of "Invadosome Formation by Lung Fibroblasts in Idiopathic Pulmonary Fibrosis"

_ijms, 2022, doi:10.3390/ijms24010499_

Round 1
Reviewer 1 Report
One minor comment:
Line 311 – 313. The authors discuss the efficacy of the inhibitors nintedanib and pirfenidone on invadosome formation and state that nintedanib is the more effective inhibitor. There are two different ways to define efficacy: 1) maximum efficacy in the given assay can be determined by conducting concentration-response experiments till maximum efficacy is reached, 2) assumed maximal clinical efficacy can be determined by using the maximum concentration that is achieved clinically. Neither 1) nor 2) was done here. A critical appraisal of the maximum clinical concentrations should be given in the discussion to relate the concentrations used in this study to the potential contribution of this effect to the clinical efficacy of the drug. For nintedanib roughly 3 -12-fold above the clinical concentration (Cmax, 40-70 nM) and for pirfenidone roughly 30 to 60-fold above the clinical concentration (Cmax, 80 µM).
Some spell ckecking required:
Abstract, line 28 …..severity of fibrosis…..
Results line 102 …. was strongly increased in IPF fibroblastic foci compared to IPF the none-fibrotic alveolar septae…
Discussion line 259: …. the first time that IPF fibroblasts has have an increased capacity to form invadosomes….
Line 280 …. the ability of isolated FPI IPF fibroblasts to degrade……
Line 307-308 …. that the marked 307 increase in invadosome formation…
Line 315 …. which are known to participate in the initiation step of invadosome formation [27]….
Reviewer 2 Report
The work by Lebel et al. investigate the role of invadosomes in the development of idiopathic pulmonary fibrosis. The study is important for the fibrosis research, the methodology is clearly described, and the results are well presented and discussed. In order to consider the manuscript for publication few clarifications are required.
- Figure 2 - each value in the histograms corresponds to a single culture from healthy and IPF individual. Do authors mean "cell line"?
- Figure 5 panels A, C, D, E the control bars are missing S.D.
Reviewer 3 Report
Manuscript ID: ijms-2010746
Title: Invadosome Formation by Lung Fibroblasts in Idiopathic Pulmonary Fibrosis
The authors investigated the invadosome formation by lung fibroblasts in idiopathic pulmonary fibrosis. The study is really interesting and well explained. There are a few areas that the authors should address to ensure that the data is clear and that there is no question concerning the data.
Here are the comments/suggestions:
1. It is relevant that authors include studies that suggest the role of fibroblast in respiratory diseases. The authors should also address this point in the introduction.
2. Please indicate whether human lung fibroblasts were from treated IPF patients? It should be addressed in the Table 1. I suggest that authors add DLCO parameter in the table too.
3. It could be interesting that authors study the association between invadosome formation (variables: invadosome-forming cells, degrade area, TKS5…) by human lung fibroblasts from IPF patients with clinical parameters (smoke status, FVC…) to evaluate the possible role of invadosome in IPF.
4. The authors did not show differences in the percentage of fibroblastic and epithelial markers between control and IPF cell cultures. However, the comparison of mean fluorescence intensity (MFI) of these markers is necessary to evaluate the differences of these variables between control and IPF patients.
5. In case authors do not find differences in MFI, they should discuss why there is an increased invadosome formation in IPF fibroblast compared to control.
6. It has been reported that pro-fibrotic cytokine TGF-β induces invadosome formation in other diseases. It could be interesting add recombinant TGF-β in the cell culture fibroblast from control or IPF patients to evaluate if there are a direct effect (future perspectives) of this cytokine in the invadosome formation. The authors should discuss the possible effect of TGF-β in invadosome formation and also address this point in the discussion.
